# Social Return on Investment of Nature-Based Activities for Adults with Mental Wellbeing Challenges

**DOI:** 10.3390/ijerph20156500

**Published:** 2023-08-02

**Authors:** Ned Hartfiel, Heli Gittins, Val Morrison, Sophie Wynne-Jones, Norman Dandy, Rhiannon Tudor Edwards

**Affiliations:** 1Centre for Health Economics and Medicines Evaluation, Bangor University, Bangor LL57 2PZ, UK; r.t.edwards@bangor.ac.uk; 2School of Natural Sciences, Bangor University, Bangor LL57 2DG, UK; h.gittins@bangor.ac.uk (H.G.); s.wynne-jones@bangor.ac.uk (S.W.-J.); n.dandy@bangor.ac.uk (N.D.); 3School of Human and Behavioural Sciences, Bangor University, Bangor LL57 2AS, UK; v.morrison@bangor.ac.uk

**Keywords:** social return on investment, nature-based activities, mental wellbeing, social prescribing, wellbeing valuation

## Abstract

Increased time spent in nature can enhance physical health and mental wellbeing. The UK Government’s ‘25 Year Environment Plan’ recommends extending the health benefits of contact with nature to a wider group of people, including those with mental health challenges. This study investigated whether nature-based interventions (NBIs) for people with mild mental health challenges could generate a positive social return on investment (SROI). Between May 2017 and January 2019, 120 participants at six outdoor sites in Wales participated in a 6 to 12-week NBI, which consisted of a weekly 2- to 4-h session. Quantitative data were collected from baseline and follow-up questionnaires identifying participant demographics and measuring mental wellbeing, physical activity, self-efficacy, and social trust. Wellbeing valuation generated a range of social value ratios by applying the Housing Association Charitable Trust (HACT) Social Value Calculator (SVC 4.0) and HACT Mental Health Social Value Calculator (MHSVC 1.0). Seventy-four participants (62%) completed follow-up questionnaires at 6 months. SROI ratios were calculated using the SVC for physical activity, self-efficacy, and social trust. The MHSVC generated social value ratios for mental wellbeing. The base case results revealed a positive social value ratio for participants, ranging from British Pound Sterling (GBP) 2.57 to GBP 4.67 for every GBP 1 invested, indicating favourable outcomes in terms of value generated.

## 1. Introduction

### 1.1. Background

Scientific evidence indicates that engaging with nature improves physical health and mental wellbeing [1,2,3]. Benefits from spending time in nature include increased physical activity, reductions in stress and anxiety, improved positive mood and self-esteem, and less social isolation [4]. Connecting people with nature is an important priority within the UK Government’s ‘25 Year Environment Plan’ [5]. 

In Wales, promoting wellbeing was a key component of the Wellbeing of Future Generations (Wales) Act 2015, which directly strengthened the duties of public bodies to work together to promote wellbeing and required public bodies to take a more joined-up approach in maximising the physical and mental health of people in Wales [6]. Green social prescribing (GSP) is a non-clinical, community-led approach to supporting people with mental health challenges. It connects people with practical, social, and emotional support in their local community through nature-based interventions (NBIs). GSP can help people with a wide range of health and social challenges, from mild to long-term physical and mental health conditions, social isolation, and frequent healthcare use [1,2,4,5].

### 1.2. Coed Lleol—Small Woods Wales

Coed Lleol—Small Woods Wales, a part of the national UK charity Small Woods Association, has been running woodland health and wellbeing programmes across 11 Welsh counties since 2010. Coed Lleol—Small Woods Wales provides NBIs for adults and families, especially for people actively using mental health services, older people in sheltered housing, people with disability or long-term illness, and people recovering from domestic violence or addictions. Although programmes are open to people from all postcodes, Active Inclusion Wales Funding prioritizes provision for those in lower-income areas, targeting long-term unemployed and economically inactive adults aged over 25 years. As a result, Coed Lleol—Small Woods Wales outdoor activity sites are located in areas with high scores on the Welsh Index of Multiple Deprivation.

Coed Lleol—Small Woods Wales programmes include a variety of outdoor activities such as bushcraft, campfire cooking, woodland walks, conservation, foraging, woodland gym, and mindfulness. Programmes are frequently offered in partnership with health or social care organisations that make referrals as part of social prescribing. Participants are signposted to ‘multi-activity’ or ‘mindfulness-in-the-woods’ programmes which run from 6 to 12 weeks (Figure 1). The length of multi-activity programmes varied from 8 to 12 weeks, depending on funding availability and specifications. For example, Active Inclusion Wales funding stipulated 12-week NBI programmes. Research suggests that NBIs offered for 8–12 weeks are most effective for improving mental health outcomes [7]. However, due to available funding, mindfulness-in-the-woods was a 6-week pilot programme. The number of participants per programme ranged from a small group of 4 to a larger group of 12, with an average of 6–7 participants per programme. Multi-activity programmes consisted of a weekly 2–4 h session, split between a physical activity and a nature-based or craft-based activity. Mindfulness-in-the-woods sessions focused on skill training modelled on the Mindfulness-Based Stress Reduction Programme [8].

### 1.3. Social Return on Investment (SROI)

SROI is a pragmatic form of social cost-benefit analysis (Social CBA). Social CBA is recommended in the HM Treasury Green Book, which provides guidance on assessing the impact, including wellbeing, of programmes and projects in the UK [9,10]. Social CBA uses quantitative methods to value relevant costs and benefits. 

SROI methodology is outlined in the Cabinet Office Guide to Social Return on Investment [11]. SROI considers which outcomes are relevant and significant to stakeholders and then assigns monetary values to important outcomes which may not have market prices. The social value of relevant and significant outcomes is compared with total costs to estimate the SROI ratio.
$$\mathrm{SROI}\;\mathrm{ratio}=\frac{\mathrm{Social}\;\mathrm{value}\;\mathrm{of}\;\mathrm{stakeholder}\;\mathrm{outcomes}}{\mathrm{Cos}t\;\mathrm{of}\;\mathrm{providing}\;\mathrm{programmes}}$$

In this study, examples of relevant outcomes for participants were improvements in mental wellbeing, physical activity, self-efficacy, and social trust [8]. The quantity of relevant outcomes can be monetised using wellbeing valuation, which provides a consistent and robust method for estimating the monetary value of outcomes that do not have market values. Recommended in the HM Treasury Green Book, wellbeing valuation uses thousands of large UK national surveys to isolate specific variables and to determine the effect of those variables on a person’s wellbeing [12]. Wellbeing valuation establishes the equivalent amount of income needed to increase a person’s wellbeing by the same amount. Using wellbeing valuation in this study enabled the generation of a range of SROI ratios which compared the costs of Coed Lleol—Small Woods Wales programmes with the monetised benefits experienced by participants.

## 2. Materials and Methods

### 2.1. Study Design

The study received ethics approval from the NHS (2017/WA/0297) and Bangor University (2017–16105). After self-referral or referral from GPs or community organisations, study participants were signposted to either an 8 to 12-week multi-activity programme or a 6-week mindfulness-in-the-woods programme. Prior to beginning their programme, study participants were given a participant information sheet and asked to sign a consent form. At the start of their first outdoor activities session, participants completed a paper copy of the baseline questionnaire with a member of the research team present to answer questions and provide emotional support if a participant experienced stress when answering a question. They completed a paper copy of the follow-up questionnaire at the start of their final session of outdoor activities. The questionnaires were designed to be positively worded, brief, easy to complete, and reduce respondent burden.

### 2.2. Study Population

Study participants (n = 120) were primarily white British (98%) between the ages of 25 and 64 (79%). Most were female (55%), only 15% were employed, and more than half (52%) lived in the 50% most deprived areas in Wales. The most common mental health challenges reported at baseline were non-clinical anxiety, depression, stress, and social isolation. These challenges were not as severe as clinical mental disorders, but they could still be a source of distress and impairment.

### 2.3. Stages of SROI Analysis

The main stages of SROI analysis include identifying stakeholders, developing a logic model, evidencing outcomes, valuing outcomes, calculating costs, and estimating the SROI ratio (Cabinet Office, 2012).

#### 2.3.1. Identifying Stakeholders

Participants who directly experienced the Coed Lleol—Small Woods Wales programme were identified as the main stakeholder. Eligible participants were adults experiencing mental or physical health challenges or both. Participants were physically well enough to participate in outdoor sessions, had a mental capacity to reflect on their own wellbeing, and were able to speak Welsh or English to understand questionnaires and focus group questions. 

#### 2.3.2. Developing a Logic Model

A logic model was created to identify expected participant outcomes. The logic model illustrated the links between inputs, outputs, and outcomes (Figure 2).

#### 2.3.3. Evidencing Outcomes

Baseline questionnaires captured demographic information, reasons for referral and validated scales to assess mental wellbeing, physical activity, self-efficacy, and social trust (Table 1). The decision to measure and value these four outcomes was based on previous informal in-house evaluations of Coed Lleol—Small Woods Wales [13] and previous research evidencing the benefits of NBIs [1,4,14]. After completing their programme, participants completed a follow-up questionnaire which included the same outcome measures.

Mental wellbeing was the primary outcome and was assessed using SWEMWBS [16]. Secondary outcomes included physical activity, self-efficacy, and social trust. Further details on the methods and results of this study are reported elsewhere [17]. 

#### 2.3.4. Valuing Outcomes 

Participant outcomes were monetised using two calculators based on wellbeing valuation: Social Value Calculator (SVC 4.0) [18] and Mental Health Social Value Calculator (MHSVC 1.0) [19]. In this study, the SVC was used to monetise the outcomes of physical activity, self-efficacy, and social trust. The MHSVC was used to monetise mental wellbeing. Because the values in the SVC incorporate mental wellbeing, the two calculators are treated separately rather than as two value sets that can be combined [12] (Table 2). 

#### 2.3.5. Valuing Outcomes through the SVC

Baseline and follow-up questionnaires were compared for each participant to determine the number of participants who improved, stayed the same, or worsened for each outcome. Valuing outcomes for physical activity, self-efficacy, and social trust was performed through the SVC, which contains a repository of more than 120 methodologically consistent and robust social values. Examples of monetary values in the SVC include GBP 3537 per year for frequent mild exercise, which can be used as a proxy for physical activity; GBP 13,080 per year for feeling high confidence, which can be used as a proxy for improved self-efficacy; and GBP 3753 per year for feeling a sense of belonging, which can be used as a proxy for social trust.

After each relevant outcome was valued, SROI methodology requires that deadweight, attribution, and displacement are considered to prevent overclaiming. Deadweight reflects the possibility that a proportion of the outcomes may have happened anyway without the intervention, attribution acknowledges that a proportion of the outcomes may be attributable to factors other than the intervention, and displacement considers whether participants had to give up any other activities from which they might have benefited. 

#### 2.3.6. Valuing Outcomes through the MHSVC

The MHVC was applied to value mental wellbeing using baseline and follow-up SWEMWBS scores for each participant [12]. After a total SWEBWBS score (ranging from 7 to 35) was recorded for each participant at baseline and follow-up, a monetary value based on wellbeing valuation was assigned to each total score. The baseline monetary value was then subtracted from the follow-up monetary value for each participant. A standard deadweight percentage for health interventions was then subtracted to calculate the total social value for each participant. 

#### 2.3.7. Calculating Costs

Total costs for the intervention included administration costs and session costs. Administration costs included time for staff to monitor and evaluate Coed Lleol—Small Woods Wales programmes, and coordinate and develop referrals and partnerships. Session costs included minimum and maximum cost scenarios for two activity leaders, instructional materials, and transporting participants to and from woodland sites via minibus.

#### 2.3.8. Estimating the SROI Ratio

SROI ratios were then calculated by comparing the total costs with the monetised outcomes calculated from the SVC and the MHSVC.

## 3. Results

Of the 120 participants who completed baseline questionnaires, 74 (62%) also completed follow-up questionnaires. Most follow-up respondents (82%) attended multi-activity programmes, while 18% participated in mindfulness-in-the-woods. Follow-up completion rates were higher for mindfulness-in-the-woods (81%) than multi-activity programmes (59%).

### 3.1. Valuing Outcomes Using the SVC

A base case and a conservative case were created to generate a range of social value ratios for the outcomes of physical activity, self-efficacy, and social trust. The base case included participants who improved or worsened by ≥10% for each outcome, while the conservative case counted only those who improved or worsened by ≥20% for each outcome (Table 3).

#### Deadweight, Attribution, and Displacement

SROI methodology requires consideration for deadweight, attribution, and displacement [11]. Deadweight was considered by applying a standard deadweight percentage of 27% for health interventions [20,21]. An attribution percentage of 46% was applied based on the number of participants (33 of 72) who reported engaging in other activities (i.e., walking, gardening, volunteering, swimming, running, cycling, Pilates, going to the gym, etc.) which could have contributed to an increase in outcomes at follow-up independent of the intervention. Displacement was estimated at 5% due to participants freely choosing to participate in the programme. Attendance was not compulsory, so participants were not required to forego other activities. When deadweight, attribution, and displacement were included for the three relevant outcomes of physical activity, confidence, and social trust, the total social value per participant was GBP 2076 per participant (base case) and GBP 1382 (conservative case) (Table 4).

### 3.2. Valuing Outcomes Using the MHSVC

Using the MHSVC for baseline and follow-up SWEMWBS scores, and accounting for 27% deadweight, the total social value per participant was GBP 1998 (Table 5, Appendix A Table A1).

### 3.3. Valuing Inputs

Total cost of Coed Lleol—Small Woods Wales programmes ranged from GBP 1658 to GBP 2588 for 6-week mindfulness-in-the-woods programmes, from GBP 2210 to GBP 3450 for 8-week multi-activity programmes, and from GBP 3315 to GBP 5175 for a 12-week multi-activity programmes (Table 6). 

The total cost per participant ranged from GBP 255 to GBP 398 for 6-week mindfulness, from GBP 260 to GBP 406 for 8-week multi-activity, and from GBP 526 to GBP 821 for 12-week multi-activity. The weighted average cost per participant per programme was GBP 428 for a minimum cost scenario and GBP 669 for a maximum cost scenario (Table 7).

### 3.4. Calculating the SROI Ratio

Using the base case SVC and MHSVC, GBP 2.57 to GBP 4.67 of social value was generated for every GBP 1 invested in Coed Lleol—Small Woods Wales programmes (Table 8).

## 4. Discussion

The body of evidence regarding the physical and mental benefits of NBIs is steadily expanding, but research on their cost-effectiveness is limited. A recent scoping review found that there is potential for NBIs to be cost-effective for people with mild-to-moderate common mental health problems, but more research is needed, such as a randomized controlled trial with sufficient statistical power and follow-up, to confirm these findings [22].

While the existing evidence on the cost-effectiveness of NBIs for treating mental health challenges is currently insufficient for the National Institute of Health and Care Excellence (NICE) to conduct a meaningful assessment and provide clinical guidance, SROI analysis is increasingly being used to estimate the wider benefits of NBIs [22].

In our SROI analysis, wellbeing valuation was applied to quantify and monetise four significant participant outcomes: mental wellbeing, physical activity, self-efficacy, and social trust. The results indicated that participation in Coed Lleol—Small Woods Wales programmes generated positive SROI ratios. 

### 4.1. Comparison with Other Studies

Based on the base case SVC and MHSVC, SROI ratios for the Coed Lleol—Small Woods Wales intervention (GBP 2.57 to GBP 4.67: GBP 1) were moderately similar to SROI ratios generated in other studies of NBIs. For example, a study of Wildlife Trust volunteers participating in a 6-week nature-based programme reported social value ratios ranging between GBP 4.20 and GBP 11.94 for every GBP 1 invested [23]. This study measured and valued five relevant outcomes: mental wellbeing, physical activity, improved health, increased nature-relatedness, and volunteering. The higher ratios observed in this study can be attributed to an error in concurrently assigning value to both ‘physical activity’ and ‘improved health,’ as these outcomes are not allowed to be combined using the SVC 4.0. 

Another SROI analysis of ‘Green Gym volunteers’ reported a social value ratio of GBP 4.02 for every GBP 1 invested with outcome measures including improved physical health, reduced social isolation, and increased personal wellbeing [24]. The Green Gym intervention involved participating in a variety of conservation activities, such as managing woodlands, growing food, creating wildflower gardens, building wildlife ponds, planting trees, and making pathways. 

An SROI analysis of the Community Garden Project at Gorgie City Farm found that the project benefited a wide range of stakeholders, including volunteers, visitors, the NHS, and the local council. Outcomes for volunteers, the primary stakeholder, included improved confidence, better mental health, being more physically active, and spending more time with friends. The analysis reported that for every GBP 1 invested in the project, GBP 3.56 of social value was generated [25].

### 4.2. Strengths of This Study

Previous SROI analyses have evaluated the social value of Wildlife Trust and Green Gym volunteers. However, this was the first to estimate the social value to people who were socially prescribed to a 6- to 12-week woodland activities programme. Second, the validity of the results was strengthened from quantitative data collected from 74 participants who completed baseline and follow-up questionnaires. Third, the social value ratios calculated in this study were generated using two value sets: SVC 4.0 and MHSVC 1.0. The value sets in both calculators are derived from wellbeing valuation, a consistent and robust method recommended in HM Treasury’s Green Book (2018) for measuring social CBA. 

### 4.3. Limitations of This Study

First, the reliability of the results may have been hampered due to the lack of a control group. As a result, other factors (e.g., weather) may have influenced how participants completed baseline and follow-up questionnaires. However, the lack of a control group was mitigated by subtracting 27% deadweight when using the MHSVC and by subtracting percentages for deadweight, attribution, and displacement when using the SVC.

Second, a common issue is that researchers working with the same data may arrive at different SROI ratios [26]. Social trust, for example, could be matched in the SVC with either ‘feeling belonging to neighbourhood’ (GBP 3753 per person per year) or ‘can rely on family’ (GBP 6784 per person per year). Matching outcomes from study data with the most appropriate value in the SVC depends on the researcher’s discretion. This can introduce potential researcher bias and the likelihood that estimates of social value can be upward biased [26]. In this study, ‘feel belonging to neighbourhood’ was chosen as the proxy for ‘social trust’, which is an outcome associated with a feeling of fitting in and being connected to a group. Future research can reduce potential researcher bias by engaging a wider variety of stakeholders in the process of valuing outcomes.

Third, Coed Lleol—Small Woods Wales multi-activity programmes were not standardised. Due to the skills and abilities of available woodland activity instructors, it is likely that the multi-activity programmes varied considerably in the delivery of content and skills. This makes it difficult to determine which woodland activities were most responsible for participant outcomes. Nevertheless, the findings from all Coed Lleol—Small Woods Wales programmes evaluated in this study showed that participants experienced relevant and important benefits.

## 5. Conclusions 

The results showed that Coed Lleol—Small Woods Wales programmes generated positive social value for participants. Quantitative data from baseline and follow-up questionnaires indicated that many participants improved in mental wellbeing, physical activity, self-efficacy, and social trust. Future analyses using SROI could assess the ‘health and social care resource use’ of participants who engage in NBIs over an extended duration. If NBIs are to be embedded in the UK healthcare system, it may be necessary to determine if participation in NBIs results in less health and social care service use and generates cost-savings for the NHS and local authorities.

The SROI findings in this study support ongoing sustainable funding for NBI projects, which deliver positive therapeutic and preventative outcomes. The results suggest that NBIs can be successfully embedded within the NHS through social prescribing, enabling health professionals to refer people to local, non-clinical services that can improve physical health and enhance mental wellbeing. 

## Figures and Tables

**Figure 1 ijerph-20-06500-f001:**
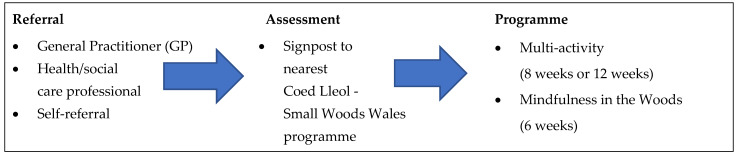
Referral process.

**Figure 2 ijerph-20-06500-f002:**
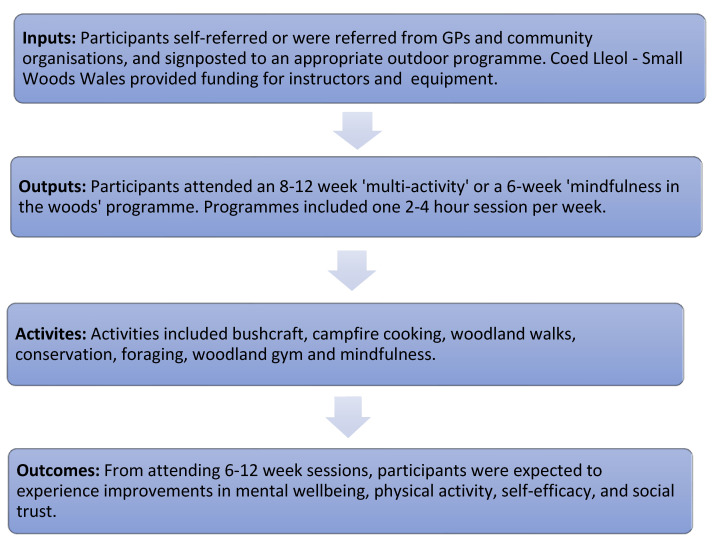
Logic Model.

**Table 1 ijerph-20-06500-t001:** Outcome measures.

Outcome	Outcome Measure	Description
Mental wellbeing	Short Warwick–Edinburgh Mental Wellbeing Scale (SWEMWBS)	SWEMWBS is a list of seven positively worded statements with five response categories to measure different aspects of positive mental health [15]. Overall scores range from 7 to 35.
Physical activity	7-Day Physical Activity Recall Question	Participants were asked: *‘On how many days in the past 7 days have you done at least 30 minutes of physical activity? (Only count physical activity that increased your heart and breathing rate, or that made you sweat a bit).’* Overall scores range from 0 to 7.
Self-efficacy	General Self-Efficacy Scale (GSES)	The GSES is a 10-item measure assessing the strength of one’s belief in their ability to respond to new or difficult situations [16]. Overall scores range from 10 to 40.
Social trust	New Economics Foundation (NEF) Social Trust Question	The NEF Social Trust Question is widely used to measure social trust [9]: *‘Generally speaking, would you say that most people can be trusted, or that you can’t be too careful in dealing with people? Please give a score of 0 to 10, where 0 means you can’t be too careful and 10 means that most people can be trusted’.* Overall scores can range from 0 to 10.

**Table 2 ijerph-20-06500-t002:** SVC and MHSVC.

Outcome	Outcome Measure	Value Set
Mental wellbeing	SWEMWBS	MHSVC
Physical activity	7 Day Physical Activity Recall	SVC
Self-efficacy/confidence	General Self-Efficacy Scale	SVC
Social trust	Social Trust Question	SVC

**Table 3 ijerph-20-06500-t003:** Base case and conservative case.

Outcome	Base Case	Conservative Case	Explanation
Physical activity (net increase)	Measured by a change of 1 point or more (≥14%) on the physical activity question (n = 24)	Measured by a change of 2 points or more (≥28%) on the physical activity question (n = 11)	The value for improved physical activity is GPB 3537 per person per year, which is the SVB value for ‘frequent mild exercise’. When the number of participants who decreased in physical activity is subtracted from the number who increased, the net increase is 24 (base case) and 11 (conservative case).
Self-efficacy (net increase)	Measured by a change of 3 points or more (≥10%) on the self-efficacy scale (n = 13)	Measured by a change of 6 points or more (≥20%) on the self-efficacy scale (n = 4)	The value for improved confidence is GBP 13,080 per person per year, which is the SVB value for ‘high confidence’. When the number of participants who decreased in self-efficacy is subtracted from the number who improved, the net increase is 13 (base case) and 4 (conservative case).
Social trust (net increase)	Measured by a change of 1 point or more (≥10%) on the social trust question (n = 18)	More social trust measured by a change of 2 points or more (≥20%) on the social trust question (n = 13)	The value for improved social trust is GBP 3753 per person per year, which is the SVB value for ‘feeling a sense of belonging to neighbourhood’. When the number of participants who decreased in social trust is subtracted from the number who improved, the net increase is 18 (base case) and 13 (conservative case).

**Table 4 ijerph-20-06500-t004:** Social value using the SVC.

Outcome	Cost Scenario	Net Quantity ^1^	Social Value Bank	Social Value Per Year	Deadweight	Attribution	Displacement	Social Value per Person
Physical activity	Base case	24/72 (33%) improved	GBP 3537 per year—frequent mild exercise	GBP 84,888	27%	46%	5%	GBP 442
Physical activity	Conservative case	11/72 (15%) improved	GBP 3537 per year—frequent mild exercise	GBP 38,907	27%	46%	5%	GBP 202
Confidence	Base case	13/69 (19%) improved	GBP 13,080 per year—feeling high confidence	GBP 170,040	27%	46%	5%	GBP 923
Confidence	Conservative case	4/69 (6%) improved	GBP 13,080 per year—feeling high confidence	GBP 52,320	27%	46%	5%	GBP 284
Social trust	Base case	18/71 (25%) improved	GBP 3753 per year—sense of belonging	GBP 67,554	27%	46%	5%	GBP 356
Social trust	Conservative case	13/71 (18%) improved	GBP 3753 per year—sense of belonging	GBP 48,789	27%	46%	5%	GBP 257
Total social value per participant (base case)								GBP 1721
Total social value per participant (conservative case)								GBP 743

^1^ Although 74 participants completed follow-up questionnaires, the total number of completed questionnaires for each outcome varied between 69 and 72 due to a small percentage of missing data.

**Table 5 ijerph-20-06500-t005:** Social value using the MHSVC.

Participants	Mean Baseline (T1)		GBP Value	Mean Follow-up (T2)	GBP Value	Difference (T2–T1)	Social Value after Deadweight (27%)
n = 70	22.6		GBP 1,350,014	24.8	GBP 1,541,561	GBP 191,547	GBP 139,829
	Total social value per participant						GBP 1998 per participant

**Table 6 ijerph-20-06500-t006:** Total costs per programme.

Cost Category	6-Week Mindfulness	8-Week Multi-Activity	12-Week Multi-Activity
**Admin costs** 1 day per week—project officer 1/4 day per week—manager	**GBP 788** (GBP 525 per mo × 1.5 mos)	**GBP 1050** (GBP 525 per mo × 2 mos)	**GBP 1575** (GBP 525 per mo for 3 mos)
**Session costs** (Minimum cost scenario)	**GBP 870** (GBP 145 per wk for 2 leaders)	**GBP 1160** (GBP 145 per wk for 2 leaders)	**GBP 1740** (GBP 145 per wk for 2 leaders)
**Session costs** (Maximum cost scenario)	**GBP 1800** (GBP 300 per wk for 2 leaders, transport, materials)	**GBP 2400** (GBP 300 per wk for 2 leaders, transport, materials)	**GBP 3600** (GBP 300 per wk for 2 leaders, transport, materials)
**Total cost per programme** (Minimum cost scenario)	**GBP 1658**	**GBP 2210**	**GBP 3315**
**Total cost per programme** (Maximum cost scenario)	**GBP 2588**	**GBP 3450**	**GBP 5175**

**Table 7 ijerph-20-06500-t007:** Total costs per participant.

Participant Categories	6-Week Mindfulness	8-Week Multi-Activity	12-Week Multi-Activity	Totals
Participants completing baseline and end-programme questionnaire	13	17	44	74
Number of programmes offered	2	2	7	11
Average weighted number of participants per programme	6.5	8.5	6.3	6.7
Average weighted cost per participant (minimum cost scenario)	GBP 255	GBP 260	GBP 526	GBP 428
Average weighted cost per participant (maximum cost scenario)	GBP 398	GBP 406	GBP 821	GBP 669

**Table 8 ijerph-20-06500-t008:** SROI Ratios.

	SVC (Base Case)	SVC (Conservative Case)	MHSVC
Total social value per participant	GBP 1721	GBP 743	GBP 1998
Total cost per participant (minimum cost scenario)	GBP 428	GBP 428	GBP 428
Total cost per participant (maximum cost scenario)	GBP 669	GBP 669	GBP 669
SROI ratio (minimum cost scenario)	GBP 4.02: GBP 1	GBP 1.74: GBP 1	GBP 4.67: GBP 1
SROI ratio (maximum cost scenario)	GBP 2.57: GBP 1	GBP 1.11: GBP 1	GBP 2.99: GBP 1

## Data Availability

The data presented in this study are available on request from the corresponding author.

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
