# Peer review of "Social Return on Investment of Nature-Based Activities for Adults with Mental Wellbeing Challenges"

_ijerph, 2023, doi:10.3390/ijerph20156500_

Round 1
Reviewer 1 Report
The authors have written a paper on SROI of nature based activities for those facing mental wellbeing challenges. The paper reads well and is presented well. It will be of interest to many involved in evaluating nature based interventions. However, some changes would help put the paper in terms of context and understandability.
Specific comments
Abstract. I presume that the numbers after (SVC) and (MHSVC) are numbers for the software versions the authors are using. I thought the numbers would be better if they were incorporated into the brackets e.g. (SCV 4.0)
Lines 39-43. It is not clear whether the 2014 and 2015 Acts are directly linked or one subsumes the other or how complementary they are. Can this be made a bit clearer.
Line 44 there is passing reference to green social prescribing in the introduction, it would be good to see more on this in terms of the existing literature. Some of the audience may not be used to what this is or what it means or how it plays out in a UK or Welsh context.
Line 45 green social prescribing is not just about referring people with mental health challenges it also includes physical and social. I think this could be made clearer when you give more context and background to what green social prescribing is.
Line 52 can you explain how the AWW programme made provision for those in lower income areas? How was this done in practise?
Line 59 is it the interest of the individuals in the groups or do the groups make a collective decision. Please also explore more about why in Figure 1 the multi-activity is 8 weeks or 12 and the mindfulness is 6 weeks. Is this to do with the activities that people can do, the activity leaders, is there any evidence about why 6, 8 or 12 weeks is good or if longer engagement i.e. 12 weeks is better than 6 for example.
2.2.1 identifying stakeholders
Don't stakeholders or participants often get involved in identifying the theory of change. Did that happen here as in 2.2.2 you make it sounds as though they were involved?
Figure 2 this theory of change could be improved, it doesn't say anything about the activities, the inputs, whether the outcomes are short, medium or long term.
Table 1 please write out NEF in full
Did participants fill in paper copy of the survey or on a tablet. How much information was given to them before the day and on the day about the study.
Line 168 were participants from urban or rural locations, how did they travel to the sites (can we get a map with the different sites across Wales, what mental wellbeing issues were they facing. What were the follow up rates for the mindfulness vs the mutli-activity.
Line 178 why was self efficacy measured by a change of 3 points or more rather than 2 or more points for p.a. and trust?
Table 5 is quite unwieldy, I suggest creating a smaller table with reduced information and placing the full table in an appendix or as supplementary information.
Line 201 is there no attribution or displacement for the MHSVC value?
Table 7 there were 7 programmes offered for the 12 week multi-activities is this because it was more popular or something else? Can you say something in the text somewhere about what is the ideal number for these activities is there a min and max. i.e. they don't run if there is less than 3 people and they can't deal with more than 8 for example.
Discussion - comparison with other studies. Is it worth mentioning Quality adjusted life years here?
Line 258 you say researchers might arrive at different SROI ratios as they choose different measures, might this be were getting input from stakeholders about what they perceive as most appropriate for the programme they have helped design - would be useful?
Discussion
It would be really useful to have a section on how the participants found filling in the different questions particularly SWEMWBS and GSES as they have multiple items. I think this would be of interest to the audience. Particularly can you reflect on this when participants have to fill it in at the beginning of their programme as they have mental wellbeing issues and some of the questions might feel quite confronted by answering them or feel much worse if they have to tick the 'none of the time' box.
Reviewer 2 Report
I understand the targeted sample is for people with mild mental challenges. So can you explain what means mild mental challenges? How did you define this group? And what does mild exercise mean?
Please include an explanation about this selected sample group in the methodology section.
Reviewer 3 Report
Review of
Social return on investment of nature-based activities for adults with mental well-being challenges
For the
International Journal of Environmental Research and Public Health
The paper investigates the social return on investment as expressed by people with mild mental challenges who participated in nature-based interventions. Social return on investment seems a quite novel method for evaluating the social return of an intervention in terms of monetary value. This approach can be quite interesting in the field of public health, as it can help persuade stakeholders and policy makers. Due to the variables observed in this study, i.e., nature-based intervention and social return of investment, I think the paper suits perfectly for the International Journal of Environmental Research and Public Health. In addition, the study is well conducted and the manuscript is well-written. I am, therefore, favourable to the publication of this paper on the Journal in its current form, as I think it well presents a Welsh (and British) project to the International scientific community.
I am also grateful to the Authors of the paper and to the Journal’s Editors for giving me the opportunity to read and review this paper. Below I reported some very minor comments, that do not add value to the manuscript. The authors can freely decide if considering or rejecting these suggestions.
Minor comments:
Line 58, Page 2. In my opinion it would read better “6 to 12 weeks”. The APA 7th suggests: “In general, use numerals to express numbers 10 and above and words to express numbers below 10. Consider on a case-by-case basis whether to follow the general guideline or if an exception applies”. In this case, I would apply an exception, as the number 6 is surely read together with 12. Not a big issue, of course.
Lines 92-93. As above, I would 8 to 12-week programme and 6-week programme. Just my opinion.
If you agree with this change, please, check the rest of the manuscript for consistency.
Line 108. I would add a reference after “A Theory of Change model …”, e.g., “H. Clark & D. Taplin (2012). Theory of Change Basics: A Primer on Theory of Change. Actknowledge” or another reference that the authors used.
Figure 2. I would indent the Figure to be aligned with the text.
Table 4. Uniform the characters’ size.
Table 5. Apply the symbol £ to all values.
References. I know the IJERPH now accepts free format submissions and states: “Your references may be in any style, provided that you use the consistent formatting throughout. [...]. DOI numbers (Digital Object Identifier) are not mandatory but highly encouraged”. However, in the present manuscript, the reference style does not seem to be aligned with any reference style guideline. The DOI or retrieval urls are rarely provided and some information seems missing, e.g., for Stewart-Brown et al. (2009) the volume, issue and pages of the article are not reported. In any case, I presume the authors will be requested to format the manuscript according to the Journal guidelines before publication.
